# Assessment of Malnutrition among Children Presenting in a Nutrition Center in Gimbichu, Ethiopia

**DOI:** 10.3390/children10040627

**Published:** 2023-03-27

**Authors:** M. Ascensión Olcina Simón, Jose M. Soriano, María Morales-Suarez-Varela

**Affiliations:** 1Direction Department, MOS Solidaria, 46400 Cullera, Spain; 2Observatory of Nutrition and Food Safety for Developing Countries, Food & Health Lab, Institute of Materials Science, University of Valencia, 46980 Paterna, Spain; 3Joint Research Unit on Endocrinology, Nutrition and Clinical Dietetics, University of Valencia-Health Research Institute La Fe, 46026 Valencia, Spain; 4Unit of Preventive Medicine and Public Health, Department of Preventive Medicine and Public Health, Food Sciences, Toxicology and Forensic Medicine, Faculty of Pharmacy, University of Valencia, 46100 Burjassot, Spain; 5CIBER in Epidemiology and Public Health (CIBERESP), Institute of Health Carlos III, Avda. Monforte de Lemos 3-5, Pabellón 11, Planta 0, 28029 Madrid, Spain

**Keywords:** malnutrition, children, Ethiopia

## Abstract

Nowadays, Ethiopia has several problems affecting children below 5 years of age, resulting in low life expectancies. Our group carried out a study to calculate the presence of malnutrition as wasting, stunting underweight, and BMI-for-age in children presenting in a nutrition center in a rural Ethiopian village in the Oromia region according to WHO guidelines. Our results demonstrated that they had moderate chronic malnutrition or stunting from 1 to 2 years of age, affecting their life, their parents, their community/household, and their country. In our viewpoint, the solution for this situation will require a global focus on several levels, including individual, family, community, and country, the last being with the help of new health policies focused on short-, medium-, and long-term strategies with multi- and interdisciplinary approaches.

## 1. Introduction

According to the United Nations, the Human Development Index (HDI), as an effective tool that includes educational care, income level, and life expectancy, helps to measure not only income, but life quality as well, with Ethiopia having the value of 0.597 in 2021 in comparison with the world average, situated at 0.721 [1]. Ethiopia is divided into eleven National Regional States (NRS) (Afar, Amhara, Benishangul-Gumuz, Gambella, Harari, Oromia, Sidama, Somali, South West Ethiopia Peoples’ Region, Southern Nations, Nationalities, and Peoples’ Region and Tigray) and two administrative cities (Addis Ababa and Dire Dawa), with this country having the thirteenth largest population in the world at 120,283,026 people and the second largest in population in Africa [2]. Approximately 22% of the Ethiopian population lives in an urban setting while 78% reside in a rural setting [3]. Several problems have been observed in Ethiopia, including (i) states’ agricultural practices, with 1.1% permanent crops out of 15.2% arable land, (ii) small farm sizes of 0.9 hectares (2.22 acres), (iii) a high fertility rate at 4.15 births per woman, and (iv) food insecurity as massive problems in the country, among others. The Ethiopian crops are based on subsistence farms where cereals (teff, wheat, maize, sorghum, and barley), coffee, potatoes, and sugarcane are grown. On the other hand, pastoral nomadism is carried out in the country with a relatively high livestock population comprising mostly cattle, sheep, goats, camel, and chickens, but they cannot afford the meat produced by the slaughtering process, as this product, together with dairy products and eggs, are sold to those of higher wealth in urban cities [4]. Getahum et al. [5] indicated that low and inadequate food consumption, associated with low agricultural production and a high prevalence of diseases, contribute to the increase in malnutrition in the country. Furthermore, armed conflict [6] in the Tigray region situated in northern Ethiopia has resulted in a blockade at all levels of humanitarian aid, along with the destruction of health infrastructure, the rampant destruction of health facilities, and the displacement of many people, being other key factors in the increase in malnutrition. In Ethiopia, 28% of child deaths are associated with undernutrition, 36% of children under five years are underweight, 23.6% are stunting and 9.9% are wasting, 22% of women aged 15–49 years are undernourished, 58% of children are exclusively breastfed during the first 6 months, 14% of children aged from 6 to 23 months are fed four or more food groups, and 45% of children aged 6–23 are fed at least three times a day [7]. The aim of this study is to carry out the anthropometric assessment of children presenting in a nutrition center in a rural Ethiopian village in the Oromia NRS.

## 2. Materials and Methods

### 2.1. Participants

We carried this study out in a rural Ethiopian village 79 km from Addis Ababa called Gimbichu in the Woreda area in the Oromia NRS, and mapping of the studied place was carried out by ©OpenStreetMap contributors (https://www.openstreetmap.org (accessed on 14 February 2023)), as is demonstrated in Figure 1. Its administrative center is Chefe Donsa, where a non-governmental organization (NGO) called Missionary Community St. Paul Apostle (MCSPA) has been working for more than twenty years in this country [8] and has recently been collaborating with a Spanish NGO known as MOS SOLIDARIA (MOSS) [9]. The sample size calculator website http://www.raosoft (accessed on 19 January 2023), with an error margin and confidence level of 5% and 90%, respectively, was used, obtaining a value of 89 children for this study. Anthropometric measurements were carried out from 2017 to 2020 for 89 children (41 girls and 48 boys) from one month to three years of age who were admitted into the nutrition center, which has built wells for this village and has continuously trained farmers in the best practices in agriculture and women in hygienic sanitary procedures, adequate nutrition, and how to nourish their families properly. Age < 3 years and having completed relevant forms, including clinical signs and personal information, were the items for inclusion. The exclusion criteria were the children being ≥3 years or that they had not completed all the information. All children admitted were screened for their nutritional status, and those found to be undernourished were included in a nutritional education program after parents signed an informed consent that was verbally translated from English to Amharic by a native Ethiopian translator fluent in English and agreed to participate before the initiation of the interview. This study was accepted by located authorities, having been approved by the Ethical Committee of the University of Valencia (Register code: 1256147) and based in the fundamental principles from the Declaration of Helsinki [10]. According to the U.S. National Bioethics Advisory Commission [11] and European Commission [12], written guardian or parental consent, or oral consent from the children when literacy was a barrier for adults, was obtained in this study. On the other hand, we also used the International Compilation of Human Research Standards applied to Ethiopia based in the Proclamation 60/1999 (Section 21) and National Health Research Ethics Review Guidelines [13].

### 2.2. Anthropometric Measurements

The growth indicators (height and weight) were calculated by M.A.O.S. to measure the level of malnutrition of children presenting in nutrition center. The subjects wore no shoes and light clothes. Weight and height were measured with a Plenna scale (model MEA 07 400, Plenna^®^, São Paulo, SP, Brazil; accuracy of 100 g) and a Seca stadiometer (model 208, Seca, GmbH, Hamburg, Germany; accuracy of 0.5 cm), respectively. The WHO’s child growth standards were applied to calculate the height-for-age z-score (HAZ) and weight-for-age z-score (WAZ), distinguishing between malnourished and healthy children according to the WHO cutoffs. Acute malnutrition or wasting was determined by WHZ, chronic malnutrition or stunting by HAZ, and global malnutrition or underweight by WAZ using the WHO reference [14]. The WHO standard cutoffs reflected moderate malnutrition with z-scores from −2 SD to −3 SD in all growth indicators. Severely malnourished scores, including those severely stunted, severely underweight, severely obese, or severely wasting, were classified by z-values < −3 SD. Furthermore, our group classified the population by the BMI-for-age z-score (BAZ) with severe thinness as cutoff-points of <−3 SD, moderate thinness as ≥−3 SD to <−2 SD, normal as ≥−2 SD to ≤+1 SD, overweight as >+1 SD to ≤+2 SD, and obese as >+2 SD [15].

### 2.3. Statistical Analysis

Data are represented as frequencies and percentages for categorical variables and as mean and standard deviation for continuous variables, and results were considered statistically significant if the *p*-value < 0.05. Values for WAZ, HAZ, WHZ, and BAZ are presented as means, grouped by sex, and SDs. To test for normality, we used a Kolmogorov–Smirnov test. According to the means, the *p*-value obtained by the independent samples t-test was calculated for normal distribution, while those for quantitative and qualitative variables were obtained with the Mann–Whitney U and χ^2^ tests, respectively. We carried out these assessments with SPSS Statistics 27.0 (SPSS Inc., Chicago, IL, USA).

## 3. Results

Table 1 shows the characteristics of the children studied in which the homogeneity in relation to gender was collected; 53.9% of the children were boys compared to 46.1% of the girls, not showing statistically significant differences, which indicates that the sample was collected proportionally in terms of gender. Table 1 reflects that children between 1 and 24 months of age were evaluated, showing statistically significant differences with children between 17 and 22 months of age being more represented. However, when age was assessed in relation to gender, no significant differences were shown.

The mean and SD values of the HAZ, WAZ, WHZ, and BAZ, in months and years, are reflected in Table 2 and Table 3, respectively, for the studied children in this Ethiopian rural village, showing no statistically significant differences in any anthropometric values. Figure 2 shows the groups of age with median, maximum, and minimum values. Table 3 demonstrates that the HAZ for children ages 1 to 2 years had a mean of −2.32, indicating moderate chronic malnutrition or stunting.

Table 4 reflects no significant differences in these previous values, including body weights and body height/length, when differentiated by gender and the total studied population.

Visible manifestations of malnutrition including trophic changes on the skin or abdominal swelling were observed in the studied children.

## 4. Discussion

Our results demonstrated that only one type of malnutrition (moderate chronic malnutrition or stunting) was observed in Ethiopian children from 1 to 2 years of age who were coming to this nutrition center for the first time, which is important to assess to guarantee a good quality of life and a high chance of survival. Yimer [16] studied malnutrition in five densely populated zones of southern Ethiopia, namely, Sidama, Semen Omo, Hadiya, K.A.T., and Guraghe, observing that in the population of 850 children aged 3 to 36 months, 45% were stunted, 42% were underweight, and 12% were wasting.

For acute malnutrition or wasting, using WHZ, Abate and Belachew [17] detected that in low income/wealth groups, children were 1.73 times more likely to develop wasting in comparison with the children of higher income/wealth households. The child populations in the East and West Gojjam zones were 17.1% and 18.6% wasting, respectively [18], 9% were wasting among primary school children in Gondar town situated in the northwest of Ethiopia [19], and 7.10% of children from 6 to 59 months of age were wasting in the Amhara NRS, specifically in the Libokemkem district [20]. In Dabat district (northwest Ethiopia), the overall prevalence of wasting was 18.2% among children aged from 6 to 59 months [21], 15.3% in street children aged from 5 to 18 years in northwest Ethiopia [22], and 12.8% of overall prevalence with 5.8% severely wasted in north Wollo, Ethiopia [23]. Furthermore, at the Dabat Health and Demographic Surveillance System (HDSS) site in northwest Ethiopia, prevalence was 17.0% among children aged from 6 to 24 months [24]; in southern Ethiopia, prevalence was 14% for 30 government primary schools [25]. Prevalence was 16.8% among children under 5 years of age in Kersa, Ethiopia [26], 89.5% in rural areas [27], 12.8% with 5.8% severely wasted in north Wollo [23], and 21.6% and 13.0% of severe and moderate wasting, respectively, in eastern Ethiopia [28], among other studies.

For chronic malnutrition or stunting, HAZ results were diverse, calculated as 37.5 and 38.3% in the East and West Gojjam zones, respectively [18]; 46.1% in Gondar town [19]; 46.4% in northwest Ethiopia [22]; 43.2% for 0–59-month-old children in Mecha and Wenberma Woredas of West Gojjam Zone, northern Ethiopia [29]; 37.9% among school-age children in Mecha District, Amhara NRS, Ethiopia [30]; 39.3% among preschool children from food-secure and food-insecure households in Albuko district, northeast Ethiopia [31], 51% in 2000 and 32% in 2016 countrywide [32], 38.39% in Ethiopian under-five children [33]; 31.8% among under-five children in West Guji Zone, Oromia NRS, Ethiopia [34]; and 18% in under-five children in Ethiopia [35], among others.

For global malnutrition or underweight, the literature obtained 22.0% and 22.5% prevalence for the East and West Gojjam zones, respectively [18]; 25% of children aged 6–59 months [36]; 24% and 7% underweight and severely underweight, respectively, of children aged from 6 to 59 months [37]; while 41%, 33%, 29%, and 24% of children <5 years of age were underweight in 2000, 2005, 2011, and 2016, respectively [38]. Prevalence was 29.3% in school children in Tikur Wuha Elementary School (in northwestern Ethiopia) [39], 15.1% for children residing in a rural area of western Ethiopia for less than two years [40], and 14% in Sodo Zuria district in southern Ethiopia [41], among others.

For the BAZ, mean values in the literature reflected −0.1, −1.06, and −0.83 for Abate and Belachew [17], Desalegn et al. [42], and Mekonnen et al. [43]. This last study observed that parasitic infections such as schistosomiasis had a negative effect on the nutritional status of school-aged children. However, this situation was exacerbated in multiparasitological infections, including *E. histolytica*/*dispar*, *H. nana*, *A. lumbricoides*, *Giardia intestinalis*, *T. trichiura*, *S. stercolaris*, *E. vermicularis*, hookworms, and *Taenia* spp. [44]. Dinku et al. [45] reflected that this parameter was negatively affected by the child’s place of residence and the sex of the household head versus positively affected by the child’s age and the child’s dietary diversity score. Furthermore, another factor to take into account in malnutrition in children is the use of local alcoholic beverages among Ethiopian populations including pregnant women. Bitew et al. [46] reflected several factors associated with alcohol consumption, namely, the educational status, history of abortion, awareness about the harmful effects of alcohol, plan of pregnancy, occupation, and family social support of these women. According to this study, an overall prevalence of 39.78% was observed in Ethiopian pregnant women versus 45%, 37.1% and 45.6% reflected in the Ethiopia Demographic and Health Survey [47], Tesfaye et al. [48], and Abetew et al. [49], among others. Anteab et al. [50] observed this tendency in 40.6% and 48.3% of the studied sample of 25 to 29-year-olds and those in the third trimester of pregnancy, respectively. Addila et al. [51] indicated that alcohol consumption during pregnancy was significantly associated with Ethiopian preterm birth.

In our study, we observed the consumption of four local alcoholic drinks in the region, known as ‘Kribo’ (or ‘Keribo’), ‘Borde’, ‘Tella’ (or ‘Talla’), and ‘Areki’ (or ‘Katikala’) [52]. The first of them is a beverage classified as a non- or low-alcoholic drink fermented with lactic acid bacteria (LAB) overnight and based in barley and sugar, popular among children and adults. ‘Borde’ is a fermented beverage, mainly with maize (but also with wheat or barley) and their malts, containing yeasts and LAB in the fermentation procedure. Its use has been observed in mothers (who after giving birth are encouraged to consume it to enhance lactation) and children (who consume it as a meal replacement with blended ‘borde’ and ‘gafuma’, which is a cooked dough with the aroma of fresh bread). ‘Tella’ is produced in households as a malt beverage with barley, maize, millet, sorghum, teff, wheat, or other cereals and fermented with mainly *Saccharomyces cerevisiae* and *Lactobacillus pastorianumi*, obtaining a range from 2.8% to 5.0% (*v*/*v*) ethanol content. The fourth of these is a prepared beer-like drink which can be distilled one or two times, called ‘Terra-areki’ or ‘Dagim-areki’, respectively. The first of them is distilled from ‘Yereki-tinsis’, which is obtained by an aqueous mixture of powdered ‘Gesho’ (*Rhamnus prinoides*) leaves and powdered ‘bikil’ (malt) in a proportion of 1:2 and fermented for around five days, while ‘Dagim-areki’ is prepared in the same way as the previous beverage, except with a distillation process that can be redistilled with three volumes of ‘Terra-areki’ or, at other times, with a distillation carried out for a shorter period of time. ‘Terra-areki’ and ‘Dagim-areki’ have mean values of 34.1% and 46.6% (*v*/*v*) ethanol content, respectively.

The consumption of alcoholic beverages, including soft alcoholic drinks, have hazardous effects on the fetus [46] and in the further development of the baby [51]. For this reason, these beverages can affect the nutritional status of children due to the intake of their mothers when they were pregnant. Tafese et al. [53] observed stunting, underweight, and wasting in 42.0%, 24.6%, and 14.5%, respectively, of 12–36-month-old children in relationship with the maternal ‘Tella’ intake. Eyasu et al. [54] studied that the perceived benefits (without scientific basis) of Ethiopian women and the rest of the population were that, for example, the intake of ‘Tella’ is beneficial to ‘hydrate’ the body of pregnant women and ‘clean’ the fetus and the uterus before and during pregnancy. Debele et al. [55] suggested that there is another reason argued by the Ethiopian community and alcohol sellers, which is that it eases their parenting duty and that the introduction of consumption of these beverages in children after six months is used so that they sleep while the mothers are gone for a long trip or to the market.

Ethiopia is also affected by the double burden of malnutrition in all the population (coexisting overnutrition, including overweight and obesity, together with undernutrition, such as stunting and wasting, at country, city, community, household and individual levels), prevalent in <2% among mother–child pairs in the studies of Sahiledengle et al. [56], and it also appears among in-school adolescents from rural Ethiopia [57] and in female adolescent students in Bahir Dar City (Amhara region) [58].

We agree in full with the idea suggested by Black et al. [59] that efforts to eradicate malnutrition should be directed towards the implementation and development of multi- and intersectoral programs, including programs to address the effects of alcohol intake in children, pregnant and lactating women, and other adults. In fact, the Nursing and Sensitive Care Framework [60] requires collaborative support across sectors and requires attention to finance, policy, and governance, along with labor and stakeholder work on intervention. Furthermore, a good tool to focus this problem is the use of UNICEF’s malnutrition framework, introduced in 1990, to work on the causes of undernutrition, highlighting health and childcare and food security conditions for adequate nutritional status [61]. This can be applied in Ethiopia for adults on highly active anti-retroviral therapy in the Wolaita Sodo teaching and referral hospital in Southern Nations, Nationalities, and Peoples’ Region [62] to reduce stunting in under-five children [63], mothers, and children from 6 to 12 months of age [64]. According to Black et al. [59], it is interesting that the new focus re-envisioning UNICEF’s conceptual framework of malnutrition has expanded to survive and thrive by including safe and nutritious food, healthcare learning opportunities, safety and security, and responsive care. In fact, the cluster growth/development promote thriving, particularly if one works into consideration both neurodevelopment and early childhood nutrition [65]. This is important for the effectiveness of the treatment. Prado et al. [66] demonstrated that the use of nutritional supplements has small, positive effects on height and development, but studies accounting for responsive attention and learning have up to five times more positive effects on neurodevelopment compared with the use of nutritional supplements alone. In our viewpoint, this new UNICEF viewpoint, together with the adoption of the Sustainable Development Goals (SDGs), according to de Romana et al. [67], in particular, with SDGs that directly affect Targets 2.2 (which is focused on ending all forms of malnutrition, including achieving, by 2025, the internationally agreed-upon targets for stunting and wasting in children under 5 years of age and addressing the nutritional needs of adolescent girls, pregnant and lactating women, and older persons) and 3.2 (end preventable deaths of newborns and children under 5 years of age, with all countries aiming to reduce neo-natal mortality to at least as low as 12 per 1000 live births and under-five mortality to at least as low as 25 per 1000 live births), along with SDGs with indirect influences, including Goals 1 (to end poverty in all its forms everywhere), 4 (quality education), 5 (to achieve gender equality and empower all women and girls), 6 (to ensure access to water and sanitation for all), 8 (to promote inclusive and sustainable economic growth, employment, and decent work for all), and 17 (to revitalize the global partnership for sustainable development), should be applied in Ethiopia, focusing on the guarantee of nutritional status and quality of life for its population. Furthermore, as Romana et al. [67] literally concluded, “although great progress has been made in the past decade, there are still many gaps that hinder this effort. Given the importance of the moment, and the fragmented nature of the nutrition ecosystem, nutrition leaders must pull different voices, agendas, and organizations together behind common objectives. Vision, creativity, diplomacy, and a good understanding of political economy are needed now more than ever to navigate differences, create alignment, increase momentum, and generate impact. Meaningful progress should focus on people, especially women, adolescent girls, and children—and put them at the center of everything we do because they hold the key to ending malnutrition. Countries, donors, and partners need to be encouraged to bring clear commitments that demonstrate how they will accelerate progress against the SDG targets, with a detailed accountability framework and plans for transparently measured investments. Finally, science and academia have an important role in accelerating the progress against the SDG targets by generating evidence, especially in the area of implementation science, by providing evidence-based policy recommendations, and in building the capacity of stakeholders at country-level”. We think that it is only in this way that the malnutrition eradication process can be successful.

## 5. Conclusions

The presence of moderate chronic malnutrition is observed in Ethiopian children from 1 to 2 years of age presenting in a nutrition center located in Gimbichu in the Oromia region. The aim of this center is to improve urgently, after a first screening, their nutritional status with tailored nutritional interventions. The focus of this evaluation should be carried out grouped in three categories: (i) individual, (ii) household, and (iii) national. In fact, the work was developed from 2017 to 2020. Unfortunately, the country was unstable due to the Tigray war carried out from 3 November 2020 to the second ceasefire in November 2022. Nowadays, this situation has changed due to the signing of a peace accord on 2 November 2022, with the African Union as a mediator. It helped to start the shipment of food aid packages to this nutrition center thanks to the efforts of NGOs, which, to guarantee the survival of these children and their quality of life, seek to reduce any type of malnutrition.

## Figures and Tables

**Figure 1 children-10-00627-f001:**
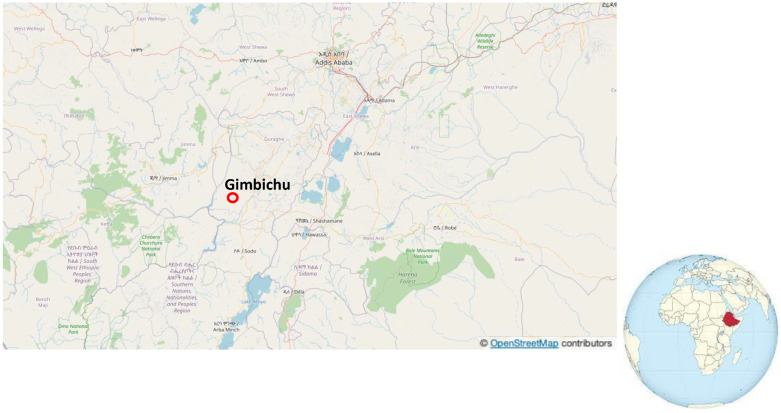
Localization of Gimbichu where this study was carried out using ©OpenStreetMap.

**Figure 2 children-10-00627-f002:**
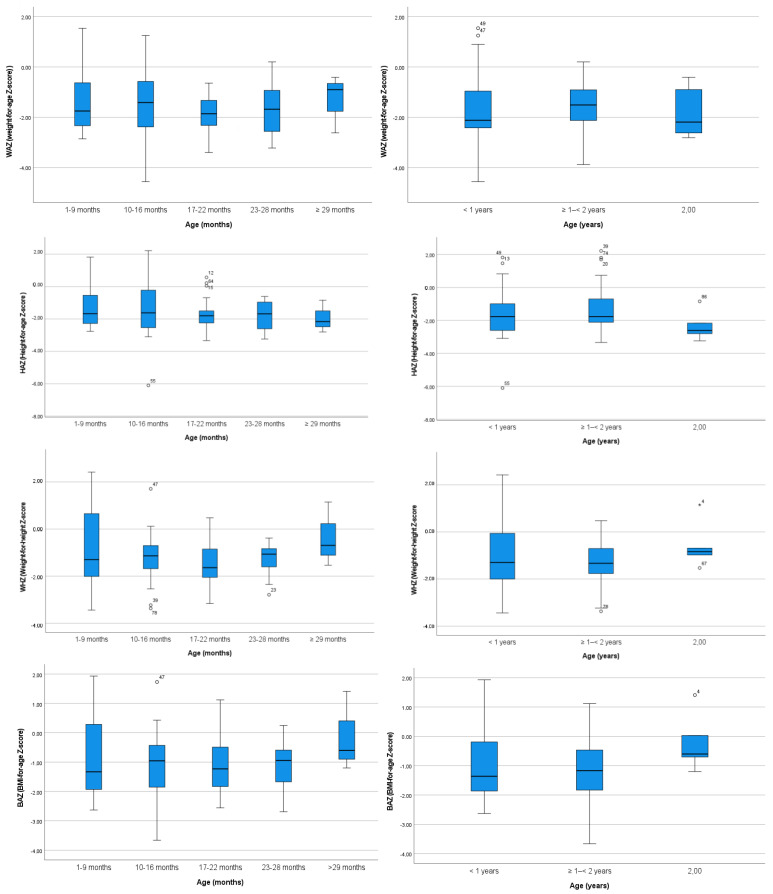
Median, maximum, and minimum values of normalized data of the WAZ, HAZ, WHZ, and BAZ for the groups of age. The circle and * are values out of range “outlier” in the distribution.

**Table 1 children-10-00627-t001:** Sociodemographic characteristics of studied children (n = 93).

Variable	n (%)	*p*-Value
Gender		
Male	48 (53.9)	0.500 *
Female	41 (46.1)	
Age (months)		
1–9 months	23 (25.8)	0.001 *
10–16 months	24 (27.0)	
17–22 months	25 (28.1)	
23–28 months	14 (15.7)	
≥29 months	3 (3.4)	
Male	15.47 ± 6.98 months	0.431 **
Female	15.29 ± 7.12 months	
Total	15.39 ± 7.01 months	

* Chi square test. ** ANOVA.

**Table 2 children-10-00627-t002:** Anthropometric values for the studied children from 1 to >29 months.

Age		HAZ	WAZ	WHZ	BAZ
1–9 months	Mean	−1.28	−1.39	−0.73	−0.82
(N = 23)	Standard deviation	1.35	1.17	1.64	1.45
10–16 months	Mean	−1.28	−1.55	−1.20	−1.12
(N = 24)	Standard deviation	1.85	1.32	1.07	1.13
17–22 months	Mean	−1.74	−1.85	−1.38	−1.08
(N = 25)	Standard deviation	0.94	0.72	0.95	1.02
23–28 months	Mean	−1.81	−1.64	−1.29	−1.06
(N = 14)	Standard deviation	0.95	1.06	0.73	0.90
>29 months	Mean	−1.93	−1.31	−0.35	−0.13
(N = 3)	Standard deviation	0.99	1.16	1.37	1.36
Total	Mean	−1.51	−1.60	−1.11	−0.13
(N = 89)	Standard deviation	1.35	1.08	1.19	1.36
*p*-value *		0.555	0.685	0.268	0.623

* ANOVA test.

**Table 3 children-10-00627-t003:** Anthropometric values for the studied children from <1 to >2 years.

Age		HAZ	WAZ	WHZ	BAZ
<1 year	Mean	−1.65	−1.65	−0.87	−0.89
(N = 33)	Standard deviation	1.50	1.31	1.52	1.36
1–2 years	Mean	−1.34	−1.55	−1.20	−1.13
(N = 51)	Standard deviation	1.26	1.55	−1.32	1.01
>2 years	Mean	−2.32	−1.78	−0.57	−0.21
(N = 5)	Standard deviation	0.91	1.07	1.01	1.00
Total	Mean	−1.51	−1.60	−1.11	−0.99
(N = 89)	Standard deviation	1.35	1.08	1.19	1.16
*p*-value *		0.234	0.846	0.142	0.199

* ANOVA test.

**Table 4 children-10-00627-t004:** Anthropometric values for the studied children distributed by gender.

Gender		Weight (kg)	Height/Length (cm)	HAZ	WAZ	WHZ	BAZ
Female	Mean	8.35	73.50	−1.17	−1.28	−0.87	−0.80
(N = 41)	Standard deviation	1.53	7.53	1.28	0.94	1.23	1.25
Male	Mean	8.19	73.12	−1.93	−1.99	−1.40	−1.22
(N = 48)	Standard deviation	1.67	8.70	1.33	1.13	1.10	1.02
Total	Mean	8.27	73.33	−1.52	−1.60	−1.12	−1.00
(N = 89)	Standard deviation	1.59	8.05	1.35	1.08	1.20	1.16

## Data Availability

The data presented in this study are available on request from the corresponding author.

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
