# Peer review of "Assessment of Malnutrition among Children Presenting in a Nutrition Center in Gimbichu, Ethiopia"

_children, 2023, doi:10.3390/children10040627_

Round 1

Reviewer 1 Report

The manuscript explored the Assessment of malnutrition among children presenting in a nutrition center in Gimbichu, Ethiopia. They showed that the presence of moderate chronic malnutrition is observed in Ethiopian children, from 1 to 2 years.

My overall evaluation of the manuscript is negative. There are a number of major revisions, formal and scientific aspects that should be addressed.

1- How to determine the sample size should be added to the article.

2- Given that the research was conducted in Ethiopia. The permission of the Ethics Committee of the Ethiopian University of Medical Sciences is also required.

3- It is necessary to conduct more studies on the amount of calories consumed and the health status of each person in order to determine how to help children. The exact type of protein, vitamin and mineral deficiency should be determined.

4- Due to the fact that children go to the nutrition center located in Gimbichu have been referred for evaluation. There is no researcher from this center among the authors of the article.

5- In the conclusion of the article, generalities are mentioned. It is necessary to specify exactly how and for how long the food aid package can be defined to solve this problem.

Author Response

Reviewer’s comment: The manuscript explored the Assessment of malnutrition among children presenting in a nutrition center in Gimbichu, Ethiopia. They showed that the presence of moderate chronic malnutrition is observed in Ethiopian children, from 1 to 2 years. My overall evaluation of the manuscript is negative. There are a number of major revisions, formal and scientific aspects that should be addressed.

Author’s comment: According to the reviewers’ comments, the manuscript has been modified. In our viewpoint, it has improved with regard to original manuscript.

Reviewer’s comment: 1- How to determine the sample size should be added to the article.

Author’s comment: According to reviewer’s comment, we have added the sample size in materials and methods section.

Reviewer’s comment: 2- Given that the research was conducted in Ethiopia. The permission of the Ethics Committee of the Ethiopian University of Medical Sciences is also required.

Author’s comment: We have not work with Ethiopian University of Medical Sciences due to that this studied center is administrated by a non-governmental organization (NGO) called Missionary Community St. Paul Apostle (MCSPA). However, several steps have carried out to guarantee ethics committee;

  • This study was accepted by the authorities of these locations, approved by the Ethical Committee of the University of Valencia (Register code: 1256147), and followed the fundamental principles from the World Medical Association Declaration of Helsinki.
  • The procedure established by the U.S. National Bioethics Advisory Commission and European Commission was used to obtain written parental or guardian consent, while oral consent was obtained from the children when literacy was a barrier for adults.
  • According to the International Compilation of Human Research Standards (see https://www.hhs.gov/sites/default/files/ohrp-international-compilation-2021-africa.pdf) , we have used two standards applied, from Ethiopian Science and Technology Commission and, Health Department, as are: i) Proclamation 60/1999, Section 21 , and ii) National Health Research Ethics Review Guideline, Fourth Edition (2014): http://www.ccghr.ca/wpcontent/uploads/2013/11/national-research-ethics-review-guidline.pdf

We have added this information in materials and methods section.

Reviewer’s comment: 3- It is necessary to conduct more studies on the amount of calories consumed and the health status of each person in order to determine how to help children. The exact type of protein, vitamin and mineral deficiency should be determined.

Author’s comment: Thank you for your comment. However, the aim of this manuscript to carry out the anthropometric assessment in children presenting in a nutrition center in a rural Ethiopian village on the Oromia region. More studies, as the suggested by the reviewer, would make in this center but the country's instability, due to Tigray war carried out from 3rd November 2020 to the second ceasefire in November 2022, did not allow it.

Reviewer’s comment: 4- Due to the fact that children go to the nutrition center located in Gimbichu have been referred for evaluation. There is no researcher from this center among the authors of the article.

Author’s comment: There is a researcher from this center among the authors of the article. M. Ascensión Olcina Simón (M.A.O.S.) carried out the anthropometric assessment in the nutrition center in Gimbichu (Ethiopia). According to the manuscript, she is working in the non-governmental organization (NGO) called MOS SOLIDARIA, which is reflected in the ‘participants section’.

In fact, we have added in the ‘Anthropometric measurements section’ that she carried out these measurements.

If you need to verify this information, you can see the work developed by this NGO in its webpage (see http://mossolidaria.org/ )

Reviewer’s comment: 5- In the conclusion of the article, generalities are mentioned. It is necessary to specify exactly how and for how long the food aid package can be defined to solve this problem.

Author’s comment: According to reviewer’s comment, this section has been modified to clarify it. In fact, the work was developed from 2017 to 2020. Unfortunately, there was a country's instability due to Tigray war carried out from 3rd November 2020 to the second ceasefire in November 2022. Nowadays, this situation changed due to sign a peace accord in 2nd November 2022, with the African Union as a mediator. It helped to start the shipment of food aid package in this nutrition center to thank the efforts of the two NGOs (Missionary Community St. Paul Apostle and MOS SOLIDARIA).

Reviewer 2 Report

The manuscript is focused on the presence assessment of child malnutrition in a nutrition center in a rural Ethiopian village in the Oromia region. The authors of the manuscript to determine it, used the indicators, there were standardized using z-scores; specifically, were used: height-for-age z-score (HAZ) and weight-for-age z-score (WAZ) and weight-for-height z-score (WHZ).   In the introduction part: - it would be appropriate to partially expand this part, as it is relatively brief. it would be appropriate to add information on the evaluation of malnourishment in children, these data are missing. - the causes of food insufficiency in the monitored area (demographic, financial, geographical, agricultural...) are very correctly described in this section.   In the material and methods section: -   it is necessary to supplement the inclusion and exclusion criteria for the selection of samples.   Results -   in table 2 (line 114) - numerical data of the main monitored indicators (HAZ; WAZ; WHZ) are missing - which needs to be corrected, as these data are the most important results of the manuscript. -   it would be enriching to state the achieved average body weights differentiated by gender (i.e., separately for girls and separately for boys) and similarly the achieved average body height/length for girls and boys. These data are missing. -   for consideration by the author of the manuscript, I give the use of the WHO categorization for parameters (WAZ; HAZ; WHZ) differentiated at the same time according to age and gender - are available categories up to 6 months of the child's age and up to 2 years of the child's age (especially for girls; especially for boys) -   it would be good to briefly expand some of the results - visible manifestations of malnutrition in children (e.g., trophic changes on the skin, abdominal swelling, what is their mental development, etc.) Discussion -   I have no comments.   Conclusion -   I have no comments.   References In the manuscript were used 36 literary sources (with 0 self-citations) of which 12 sources are from the last 5 years. Literary sources need to be updated and supplemented with newer ones.

Author Response

Reviewer’s comment: The manuscript is focused on the presence assessment of child malnutrition in a nutrition center in a rural Ethiopian village in the Oromia region. The authors of the manuscript to determine it, used the indicators, there were standardized using z-scores; specifically, were used: height-for-age z-score (HAZ) and weight-for-age z-score (WAZ) and weight-for-height z-score (WHZ).   In the introduction part: - it would be appropriate to partially expand this part, as it is relatively brief. it would be appropriate to add information on the evaluation of malnourishment in children, these data are missing. 

Author’s comment: According to your comment, we have added this information in the introduction section.

Reviewer’s comment: the causes of food insufficiency in the monitored area (demographic, financial, geographical, agricultural...) are very correctly described in this section.   

Author’s comment: Thank you for your comment.

Reviewer’s comment: In the material and methods section:   it is necessary to supplement the inclusion and exclusion criteria for the selection of samples.  

Author’s comment: According to reviewer’s comment, we have added this information in the materials and methods section.

Reviewer’s comment: Results -   in table 2 (line 114) - numerical data of the main monitored indicators (HAZ; WAZ; WHZ) are missing which needs to be corrected, as these data are the most important results of the manuscript. 

Author’s comment: According to reviewer’s comment, we have added these values and we have separated in two tables and incorporated Figure 2 to clarify it.

Reviewer’s comment:   it would be enriching to state the achieved average body weights differentiated by gender (i.e., separately for girls and separately for boys) and similarly the achieved average body height/length for girls and boys. These data are missing. 

Author’s comment: According to reviewer’s comment, we have added these values in Table 4.

Reviewer’s comment:  for consideration by the author of the manuscript, I give the use of the WHO categorization for parameters (WAZ; HAZ; WHZ) differentiated at the same time according to age and gender are available categories up to 6 months of the child's age and up to 2 years of the child's age (especially for girls; especially for boys).

Author’s comment: Thank you for your comment. We have used this information in the manuscript.

Reviewer’s comment:   it would be good to briefly expand some of the results

Author’s comment: According to reviewer’s comment, it has briefly expanded.

Reviewer’s comment: visible manifestations of malnutrition in children (e.g., trophic changes on the skin, abdominal swelling, what is their mental development, etc.) 

Author’s comment: According to reviewer’s comment, it has been added in the results section.

Reviewer’s comment: Discussion -   I have no comments.   

Author’s comment: Thanks.

Reviewer’s comment: Conclusion -   I have no comments.   

Author’s comment: Thanks.

Reviewer’s comment: References In the manuscript were used 36 literary sources (with 0 self-citations) of which 12 sources are from the last 5 years. Literary sources need to be updated and supplemented with newer ones.

Author’s comment: According to reviewer’s comment, we have added new references.

Round 2

Reviewer 1 Report

According to the explanations given by the authors of the article and the changes made in the article. The article is acceptable.